# The Role of Mesenchymal Stem Cells (MSCs) in Veterinary Medicine and Their Use in Musculoskeletal Disorders

**DOI:** 10.3390/biom11081141

**Published:** 2021-08-02

**Authors:** Przemysław Prządka, Krzysztof Buczak, Ewelina Frejlich, Ludwika Gąsior, Kamil Suliga, Zdzisław Kiełbowicz

**Affiliations:** 1Department of Surgery, Faculty of Veterinary Medicine, Wroclaw University of Environmental and Life Science, Pl. Grunwadzki 51, 50-366 Wroclaw, Poland; krzysztof.buczak@upwr.edu.pl (K.B.); zdzislaw.kielbowicz@upwr.edu.pl (Z.K.); 22nd Department of General Surgery and Surgical Oncology, Wroclaw Medical University, Borowska 213, 50-556 Wroclaw, Poland; ewelina.frejlich@umed.wroc.pl; 3Vets & Pets Veterinary Clinic, Zakladowa 11N, 50-231 Wroclaw, Poland; ludwika.gasior@gmail.com; 4Student Veterinary Surgical Society “LANCET”, Faculty of Veterinary Medicine, Wroclaw University of Environmental and Life Science, Pl. Grunwaldzki 51, 50-366 Wroclaw, Poland; 106495@student.upwr.edu.pl

**Keywords:** mesenchymal stem cells, cell therapy, regenerative medicine, diseases of the musculoskeletal system, orthopedic disease, veterinary medicine, MSC

## Abstract

Regenerative medicine is a dynamically developing field of human and veterinary medicine. The animal model was most commonly used for mesenchymal stem cells (MSCs) treatment in experimental and preclinical studies with a satisfactory therapeutic effect. Year by year, the need for alternative treatments in veterinary medicine is increasing, and other applications for promising MSCs and their biological derivatives are constantly being sought. There is also an increase in demand for other methods of treating disease states, of which the classical treatment methods did not bring the desired results. Cell therapy can be a realistic option for treating human and animal diseases in the near future and therefore additional research is needed to optimize cell origins, numbers, or application methods in order to standardize the treatment process and assess its effects. The aim of the following work was to summarize available knowledge about stem cells in veterinary medicine and their possible application in the treatment of chosen musculoskeletal disorders in dogs and horses.

## 1. Introduction

The world of science is still very much interested in the topic of stem cells. In the simplest way, they can be characterized as cells that have the ability to self-renew and differentiate into other types of cells. [1,2,3,4]. These cells are present in every living organism from the moment the ovum is fertilized until the moment of death. Their presence allows the body to develop and maintain the number of somatic cells in balance. They also enable the regeneration of organs and tissues by replacing somatic cells that deteriorate over time or are damaged [3,5,6,7].

The discovery and development of methods for obtaining stem cells allowed us to come much closer to implementing the age-old dream of mankind regarding replacing sick and worn-out cells and/or tissues with new ones, that are grown in the laboratory. The importance of stem cells in medicine was emphasized by the Nobel Prize award that acknowledged two scientists, John Gurdon and Shiny Yamanaka for the development of the so-called induced pluripotent stem cells. They are reprogrammed somatic cells that acquire the features of stem cells [8,9,10]. Thanks to numerous studies on stem cells in various fields of science, it is possible to use them in human [11] and animal medicine [12,13].

The characteristic features of stem cells mentioned by numerous authors include their simple structure and lack of differentiation; self-renewal, which allows one to maintain a constant population of cells throughout the life of an organism; asymmetrical cell division that results in the formation of a larger stem cell and a smaller cell undergoing further linear differentiation; the ability to differentiate into cells of various tissues; and the expression of proteins c specific to undifferentiated cells, i.e., c-kit, Thy1 [2,4,14,15].

This paper summarizes the currently available knowledge about stem cell use in the field of veterinary medicine. It describes various forms of stem cells, their immunomodulatory properties, and an evaluation of the therapy’s effectiveness in the treatment of joint, ligament, and tendon diseases in dogs and horses, based on selected scientific publications.

## 2. Division of MSCs

There are many types of MSCs, occurring at different places and different periods of time during the life of an organism. They vary between themselves in their proliferation potential, the ability to differentiate into other types of cells in the body, the source of their origin, and in the relation to the recipient [2,7,14,16].

Taking into account the properties of MSCs in terms of their ability to differentiate into other types of cells, we distinguish totipotent, pluripotent, multipotent, and unipotent cells [2,7,14,16]. Totipotent stem cells are cells that exhibit unlimited dividing capacity. These cells can give rise to the entire body, which is due to the ability to differentiate into any type of cells that build the embryo and extra-embryonic tissues (placenta, umbilical cord). The totipotent cell is a fertilized ovum (zygote) and cells obtained from the first germinal stage (morula), as evidenced by monozygotic twins produced from different blastomeres. Pluripotent stem cells are cells that can transform into any of the three germ layers: The endoderm, ectoderm, mesoderm, and the cells derived from them. Unlike the cells described earlier, they cannot give rise to the entire organism. Examples of pluripotent cells are the blastocyst germ cells, referred to as inner cell mass (ICM). Multipotent stem cells are cells that can differentiate into all types of cells that originate from the germ layer. Examples include hematopoietic stem cells found in the bone marrow or umbilical cord blood. Unipotent stem cells, also called precursor cells, show a targeted differentiation mechanism into a specific type of mature body cells. Under regular conditions, they allow the maintenance of a constant cell number in the tissues; an example here is the reproductive layer cells of the epidermis [2,7,14,16].

Taking into account the source of MSCs, we can distinguish embryonic, fetal, and adult types of cells. Embryonic stem cells (ESCs) obtained from the embryo of blastomeres are totipotent. On the other hand, for experimental purposes, cells from the blastocyst embryonic node, which are pluripotent, are most often used. These cells can transform into all types of cells in the organism, while at the same time showing unlimited self-renewal capacity in vitro. Fetal stem cells (FSCs) derive from fetal tissues, umbilical cord tissues (e.g., Wharton’s jelly), umbilical cord blood, and amniotic cells. They show a multipotential character. Adult stem cells (ASCs) are also known as mature or somatic. They are undifferentiated cells with multi- or unipotent properties. They occur in the body in the postnatal period, giving the possibility of tissues and organs in which they occur [2,7,14,16].

MSCs can also be divided according to its relation to the recipient, distinguishing stem cells of autogenous, allogeneic, and xenogeneic origin. Autogenous stem cells are currently the most popular among scientists and clinicians. These cells are isolated from their source from the donor patient (e.g., bone marrow, adipose tissue, cord blood) who is also the recipient. Then they get applied to the regenerating tissue or organ. This procedure aims to stimulate the repair of tissue damage by differentiating to the desired cell/tissue type [1,16]. Allogeneic stem cells are cells taken from another individual of the same species. They constitute the basis of therapy with the use of embryonic and mature stem cells. This method allows one to use an appropriate number of necessary cells without having to be bothered by time limitations [7,16,17]. This kind of cell can be obtained long before the implantation procedure and can also be multiplied and stored for quite some time. Using these cells as opposed to autogenous ones allows one to not expose the patient to additional anesthesia and trauma during the collection of cells. The limitation may be having to have a donor. Xenogeneic stem cells are an alternative to autologous and allogeneic stem cells [17,18]. Numerous studies with the use of antlerogenic (xenogenic) stem cells, marked with the symbol MIC-1, showed a regenerative effect on the tissues of other mammals, such as rabbits [19,20] and horses [21].

ESCs, given as an ideal example of stem cells in the natural environment, have unlimited in vitro potential for self-renewal and differentiation into each type of cell in an organism. At the same time, it has not been possible to develop precise procedures for managing their differentiation and division at the site of administration into the body, which may lead, among others, to the development of neoplastic lesions observed in experimental animals. That and the difficulties of ethics human medicine is why the ESCs have not yet been used in clinical stem cells therapy [7,16,22].

The discovery and development of methods of obtaining adult stem cells made it possible to significantly avoid the problems mentioned above, allowing them to be used in clinical treatment. These cells are presumed to be present in all organs of the body [15]. This is evidenced by, inter alia, their presence in tissues with negligible regenerative abilities, such as nervous tissue [23]. Despite the small number of ASCs in the adult organism, difficulties in obtaining them, and their lower ability to self-renew and differentiate compared to ESCs, these cells have attracted great interest, arousing high hopes in regenerative medicine [16].

The best-known cells that have been used for years are hematopoietic stem cells of the bone marrow, the stroma of which also contains MSCs. They show the ability to adhere to the substrate and divide with the formation of fibroblast-like cells [2,3,24,25]. These cells, with the possibility of self-renewal and multipotential differentiation, are also obtained for clinical purposes from adipose tissue [12]. Despite the basic range of MSCs differentiation into chondrocytes, osteocytes, or adipocytes [2,3], under laboratory conditions these cells can also differentiate into cells such as myocytes, hepatocytes, or neurons [6,16,26].

Autogenous stem cells of pluripotent nature can be obtained by transforming adult somatic cells, for example, fibroblasts (cultured in vitro), and are an alternative to pluripotent cells of embryonic origin. They are obtained by introducing genes encoding transcription factors necessary for the development of embryonic cells (c-Myc, Klf 4, Oct 3/4, Sox 2) [8,22]. The resulting cells can differentiate similarly to embryonic cells and are referred to as induced pluripotent stem cells (iPSCs) [10,22]. Unfortunately, the effectiveness of this procedure is low, and the cells obtained in this way, administered to laboratory animals, caused (similarly to ESC) the development of teratomas [7].

## 3. Tissue Sources of MSCs

Previous studies show that MSCs can be successfully isolated from almost any tissue of a living organism [27,28]. However, due to the proliferation and differentiation potential of these cells, the best results are obtained when collecting material from adult tissues such as bone marrow (BM-MSCs), adipose tissue (AD-MSCs), peripheral blood (PB-MSCs), or fetal tissue such as the placenta (P-MSCs), umbilical cord (UC-MSCs), or umbilical cord blood (UCB-MSCs) [27,28].

Based on the research, it was found that the type of cells, depending on the place of their origin, has a significant impact on the potential of their differentiation in vivo [29,30], as well as on their biologically significant features. Another important aspect in selecting the source of MSCs should be their use in a given therapy.

BM-MSCs are isolated directly from bone marrow aspirate. This tissue, similarly to human medicine, is one of the most studied sources of MSCs origin in veterinary medicine [31]. Unfortunately, collecting the material is associated with an invasive procedure performed under general anesthesia in dogs and sedation with or without local anesthesia in horses with the same risk of postoperative complications, such as infection or bleeding [32,33]. There have been some case reports of an accidental fatal thoracic and cardiac puncture [34] and nonfatal pneumopericardium in horses [35]. Nevertheless, in these animals, the procedure of collecting the BM-MSCs may be performed on a standing animal without much risk of the aforementioned complications. Nowadays, it is recognized that the most suitable site is sternal biopsy using a Jamshidi needle at the fourth or fifth sternebra [36,37]. It is known that BM-MSCs constitute only a small number of all bone marrow stromal mononuclear cells, and their number decreases with age [38]. To increase the number of MSCs, cultivation and their passaging in vitro from the aspirate obtained earlier from the donor are used. This culture, which can be passed for a maximum of four times, consists of several stages (establishment of the culture, cell multiplication, replacement of the medium, and completion of the culture with the determination of the phenotype of cells) and is strictly dependent on the number of MSCs in the therapeutic dose of the preparation that we want to obtain.

In recent years, adipose tissue has been an increasingly popular and more easily accessible source of material for MSCs extraction and isolation [30]. The ease of obtaining the material comes down to planned lipectomy or lipoaspiration performed during prophylactic surgery in dogs and cats, which is ovariohysterectomy. A suggested source of AD-MSCs is subcutaneous or visceral fat [39]. An insignificant number of nucleated cells containing traces of AD-MSCs can be obtained from the aspirate immediately after collection. As in the case of BM-MSCs, to obtain a therapeutic dose of fat-derived cells, it is necessary to cultivate them in vitro, preceded by specific laboratory procedures, such as enzymatic digestion, and then a series of washes and centrifugations. AD-MSCs have been shown to have a high proliferation and multidirectional differentiation capacity, making these cells a frequently used source for regenerative medicine. These cells have been and are used many times in the treatment of diseases of the musculoskeletal system, and their bioavailability and methods of being obtained compared to BM-MSCs contained in the bone marrow speak as a better source for their use [40,41,42,43].

PB-MSCs can also be considered an alternative source of MSCs. Compared to the methods described above, taking a blood sample is the least controversial method, carries the lowest risk of complications, and does not require pharmacological sedation of the animal to perform this procedure. However, current research requires further continuation due to the low bioavailability of PB-MSCs in the peripheral blood of humans, dogs, and horses. These cells were also successfully isolated from rabbits, mice, and guinea pigs [44,45]. The authors report that only 3 out of 10 test horses have enough cells in the blood for further cell culture. These animals were also subjected to specific therapies (hyperbaric chamber) to increase the number of PB-MSCs in the peripheral blood [46,47].

Another also promising and more often used source of stem cells in human and veterinary medicine is fetal tissues (amniotic fluid, fetal membranes, placenta, Wharton jelly of the umbilical cord (WJ), and cord blood) [48,49,50,51]. It is known that the performance of MSCs declines with the age of the donor. It has been shown that cells obtained from fetal tissues can develop not only mesenchymal cells but also into the cells of the other three germ layers, which may indicate their pluripotency [52]. Other frequently mentioned features of cells of embryonic origin include nonrejection by a foreign recipient (xenogeneity feature), the ability to migrate beyond the site of application, and longer survival of cells from adult donors [38,53]. Compared to the collection of BM-MSCs or AD-MSCs, the procedure for collecting the material is classified as a minimally invasive technique and takes place in the perinatal period. The mentioned disadvantage of the material packaging technique may be the low sterility of the procedure itself [54].

## 4. Autologous and Allogeneic MSCs

Nowadays, regenerative medicine uses mainly two types of stem cells—cells of allogeneic and autogenous origin. There is an ongoing debate among scientists around the world regarding whether there is a difference in the safety and efficacy of using these cells in treatment. As mentioned above, based on the donor–recipient relationship, MSCs can be classified as cells of allogeneic, autologous, or xenogeneic origin. Allogeneic cells are collected from a donor and used in the recipient of the same species. These cells can be successfully cultured in vitro, which significantly reduces the waiting time for the implementation of the therapy concerning autologous cells. Autologous cell transplant is performed in the same individual and requires postponement of the transplant procedure, which is closely related to the isolation and culture of cells. On the other hand, the least frequently used source of MSCs are xenogeneic cells, where the donor belongs to a species other than the recipient, as exemplified by MIC-1 cells. The Antlerogenic Stem Cells MIC-1 (collected from a deer) were experimentally inserted into damaged rabbits’ tissues and clinically given to horses [19,20,21,55].

In the era of regenerative medicine heyday and when the search for alternative treatments peaks, scientists and clinicians have been trying to prove which cells are better—allogeneic or autologous. Before deciding on the type of therapy one wants to use, one must consider the advantages and disadvantages of each type of MSCs. Logic dictates that the safest source of MSCs for an individual is cells taken and isolated directly from him. It is believed that the use of autologous cells from the patient for treatment is safer, but on the other hand, it is associated with a long procedure of obtaining (often a surgical procedure) and preparing a therapeutic dose of MSCs (isolation and cell culture). The above-mentioned problems and the cost of the whole procedure are why this approach is used in a limited manner. The number and quality of autologous cells in a particular patient fluctuates and is also closely related to their age, sex, and type of illnesses [26,56,57,58].

Cells of allogeneic nature after donation can be grown and kept in a cell bank indefinitely. In this way, a specific cell batch with a constant MSCs number, safety level, and potential differentiation can be determined. Such an approach could reduce the typical variability, as in the case of autologous cells, allowing for homogeneity of the therapy and the expected results, and would also save the time needed for material collection and cell culture [57,59]. In the case of the use of allogeneic cells, the legal regulations in force in the country where the medicinal preparation is to be used should also be taken into account. Interesting test results were presented by Bertoni et al. [60] after they evaluated the effect of auto- and allogeneic stem cells on healthy fetlocks. The cited authors demonstrated that the marrow-derived MSCs induced significantly more synovial effusion compared to umbilical cord blood-derived MSCs, although no significant difference was noted within the synovial fluid parameters. What is interesting, however, is that mesenchymal stem cell injections induced mild to moderate local inflammatory signs compared to the placebo, wherein a larger number of cells displayed a lesser inflammatory reaction in clinical and ultrasonographic exams [60]. Many experimental works using stem cells of xenogeneic origin have been published. These cells were administered both locally and systemically (intravenously) using various experimental models: Mice, rats, rabbits, dogs, and baboons [17,61]. Studies performed on rats showed the presence of xenogeneic stem cells (murine) in the bone marrow 12 weeks after their administration to the carotid artery. Their presence, apart from the bone marrow, has also been demonstrated in the tissue of myocardial infarction, where they were observed in the form of immature myocytes [62]. Successful attempts to use xenogeneic stem cells have allowed for further research, such as the use of rat stem cells in the regeneration of bone tissue in rabbits [63], human bone marrow stem cells in the treatment of spinal cord injuries in rats [64], and the same cells in bone regeneration in mice [65].

Confirmation of allogeneic stem cells tolerance is two allogeneic cell therapy products in horses and one in dogs that have a marketing authorization of the European Medicines Agency, among others. Analyzing currently available information, an increasing number of researchers suggest that the use of allogeneic or autogenous stem cells in regenerative medicine can be replaced with cells of xenogeneic origin. Studies clearly show that the immunogenicity of xenogeneic stem cells is similar to those in the auto- and allogeneic configuration [17,18,19].

## 5. Immunumodulatory Properties of MSCs

Normal allogeneic cells or organs would be rejected by an immune response. This happens because of the presence of a major histocompatibility complex (MHC) on the surface of living organism cells. MHCs are a class of molecules that influence the capability of an organism to accept or reject transplanted tissues. Class I MHC molecules span the membrane of almost every nucleated cell in an organism and class II molecules are concerned with cells of the immune system known as macrophages, dendritic cells, and lymphocytes [66]. This means that every allogeneic cell with MHC I surface molecules is recognized by recipient CD8+T cells, leading to direct cytotoxicity of foreign cells. If they have MHC II molecules, they can be recognized by recipient CD4+ T cells, leading to either cytotoxic or humoral immune response. In addition, B cells could also produce alloantibody after indirect recognition by antigen presenting cells [29,67].

Despite the lack of precise knowledge of the immunogenicity of adult MSCs, they are considered to be hypo immunogenic. Among other things, this is due to the lack of histocompatibility MHC class II antigens on their surface and the poor expression of MHC class I. They inhibit the proliferation and function of T and B lymphocytes and NK cells. They also inhibit antigen-presenting cells and stimulate the proliferation of suppressor T cells [68,69,70,71]. After the MSCs induction to differentiate along the adipogenic, osteogenic, and chondrogenic lineages, they express MHC class I but not MHC class II molecules on their surface [72]. This phenotype observed in undifferentiated and differentiated MSCs (along the adipogenic, osteogenic, and chondrogenic lineages) is considered non-immunogenic and suggests that these cells can induce tolerance [70,72]. Research conducted around the world has shown that stem cells also exhibit the same immunomodulatory properties both in allogeneic and xenogeneic recipients [17]. A fine example of these features is the research using green fluorescent protein (GFP) labeled autologous and allogeneic mesenchymal progenitor cells (MPCs) that were injected into artificially made superficial digital flexor tendon (SDFT) lesions in horses. The results showed no differences in either the number or distribution of autologous and allogeneic cells at the respective injection sites. There were no external or histological signs of increased inflammation compared to the autologous injection site [73]. Because the MSCs seem to be immune privileged, the allogeneic type could possibly provide a readily available source for regenerative medicine in veterinary medicine. Nevertheless, the risk of disease transmission from the donor to the recipient has to be kept in mind [74]. For this reason, it is important to underline the requirement to schedule a specific systemic plan of control of potential viral, bacterial, and fungal contamination for each species.

## 6. Application of MSCs in Veterinary Medicine–Musculoskeletal System Disorders

The possibility of using stem cells raises great hopes in the treatment of, among others, chronic and degenerative diseases or damage to organs and tissues [1,4]. Currently, MSCs from bone marrow or adipose tissue are mostly used clinically in veterinary medicine. Their therapeutic applications in animals include, in dogs and horses, the treatment of the musculoskeletal system (tendons, ligaments, and joints) diseases. Additionally, MSCs are being used in recurrent airway obstruction in horses [75], while in cats, there are attempts to use them in digestive system disease (inflammatory bowel disease) and chronic kidney disease [76,77]. Experimentally in mice, rats, and dogs, possibilities of using MSCs to treat acute liver failure were evaluated [78,79,80,81,82]. Moreover, auto- and allogeneic stem cells have been used in the experimental treatment of spinal cord injuries [26,83], induced urinary incontinence [84], mucosal ulcerations [85], muscular dystrophies [86], bone defects [87], articular cartilage [88], and diabetes [89].

Musculoskeletal disorders currently account for the highest percentage of clinical cases in veterinary medicine for which stem cells have been used. These diseases may involve not only the articular apparatus, but also muscles, ligaments, tendons, and bones. Depending on the nature of the disease there is a wide range of symptoms, such as inflammation and pain in a given area of the body, lameness of varying degrees, and reluctance to undertake physical activity by the animal. The most common diseases of the musculoskeletal system include osteoarthritis (OA), tendon and ligament injury (mainly in horses, TLI), and intervertebral disc disease (IVDD). So far, the treatment of these diseases has included surgical techniques, systemic and local application of anti-inflammatory preparations, hyaluronic acid (HA), specialized cellular products (platelet-rich plasma—PRP, interleukin-1 receptor antagonist protein—IRAP), restriction of movement, and physiotherapy [90,91,92,93]. For several years, regenerative medicine has been offering MSCs therapies alternative to traditional treatment methods [13,31,40,41,42,43,93,94,95,96,97,98,99,100,101,102,103,104,105,106,107,108,109].

### 6.1. Degenerative Joint Disease

OA is a painful and disabling disease that is an increasing problem in both animal and human populations [110,111,112]. Currently, a significant number of young patients are consulted for joint problems. The poor cell matrix of the articular cartilage and the lack of penetrating blood vessels, lymphatic vessels, or nerves do not favor the processes of repairing its damage [13,94,113,114]. This reparation is possible only during the early development of the organism. The defect is usually filled with fibrous or hyaline-like cartilage, which has worse biomechanical properties. OA develops and progresses with age and is manifested by inflammation and successive degeneration of cartilage and subchondral bone, resulting in deterioration and loss of full functionality of the joint. During movement or in animals taking an active part in sport, this tissue is subjected to significant loads resulting in microinjuries. Lameness caused by advanced degenerative disease is quite a common condition of racehorses, leading to complete exclusion from the sport. This plays a significant economic role for the owners of these animals [93,115]. This disease is often accompanied by various stages of developing joint dysplasia in dogs or cats.

In recent years, stem cells have been used many times as an alternative healing therapy in dogs and horses. HA, PRP, or other medicinal preparations were often used with MSCs at the same time. Selected animal studies describing the use of MSCs in the treatment of OA in dogs and horses are summarized in Table 1. In all cases, the selected cells were administered directly as an intra-articular injection, acupuncture point injections, or intravenously. An honest evaluation of MSCs activity and their effectiveness in treating OA is possible only when the results can be compared with a control group. However, based on the available literature, it seems that it is not always possible, mainly concerning presentations of clinical patient studies [13,95,97,102]. Studies with a parallel control group allow one to draw clear conclusions from the experiment and correctly evaluate implemented treatment in clinical patients. A randomized, blinded, placebo-controlled clinical trial reporting on the effectiveness of stem cell therapy in dogs showed that injecting the AD-MSCs into the hip joint leads to significant improvement, which manifests in less lameness, less pain, and greater range of motion throughout the clinical trial in comparison with a control group [40]. Other similar studies have shown that MSCs and plasma rich in growth factors (PRGF) in a control group provided a significant improvement, reducing dogs’ pain and enhancing physical function. At the same time, with respect to basal levels for every parameter in patients with hip OA, MSCs showed better results at 6 months [41]. This is also confirmed by studies in dogs, using AD-MSCs linked with PRGF, where an increase in functionality and less discomfort associated with pain sensation caused by OA of the hip has been shown [43]. The effectiveness of stem cells in treating OA in dogs was also evaluated when the elbow and stifle joints were involved [13,94,95,99,100]. In the treatment of experimentally induced OA of the stifle joint in dogs, the group with joined AD-MSCs with platelet-rich plasma (PRP) given intraarticular was advantageous over the control group (phosphate buffered saline) and over sole injections of PRP or AD-MSCs. A beneficial effect of the medicinal preparations was found, demonstrating their stimulation for the synthesis of the extracellular matrix (ECM) and the proliferation of chondrocytes. These preparations also inhibited the inflammatory reaction within the joint [100]. Intraarticular application of AD-MSCs when treating dogs with chronic osteoarthritis of the humeroradial joints showed a significant improvement compared with pharmacological therapy used earlier. Features of canine umbilical cord mesenchymal stem cells (UC-MSCs) in treating OA were studied both experimentally [99] as well as in clinical patients [94]. Experiments on dogs showed significantly higher improvement in stifle joint cartilage neogenesis and recovery in the treated group (UC-MSCs) compared to the untreated control group (saline). In the treated group, the joint fluid and the inflammatory response decreased [99]. Likewise, no adverse effects associated with UC-MSCs were noted in clinical patients with OA of the elbow joint. Additionally, on the basis of six-month observation, clinical improvement was noted in dogs after injecting the UMSCs intra-articullary in comparison to dogs in the control group, injected with saline [94]. Possibilities of using stem cells were also examined in treating horses with OA, both in auto- and in allogeneic systems [93,96,98]. Interestingly studies have evaluated the effectiveness of the treatment when autological BM-MSCs are given with hyaluronan (HA) in comparison to giving HA only to the stifle joint with an experimentally made cavity in the joint cartilage. The cited authors showed no significant clinical or histological difference in the two groups; however, their results confirm that intra-articular BM-MSCs enhance cartilage repair quality with increased aggrecan content and tissue firmness [98]. Clinical studies evaluating the possibility of using veterinary product consisting of allogeneic chondrogenic induced mesenchymal stem cells (ciMSCs) with equine allogeneic plasma showed superior efficacy in comparison to the control group (saline). Clinical improvement was observed even one year after administration, with significantly more horses working at their previous level or at their training level at this point compared to the placebo control group [96]. Magri et al. [93] compared the efficacy and safety of single versus repeated intra-articular injections of allogeneic mesenchymal stem cells for the treatment of osteoarthritis of the metacarpophalangeal/metatarsophalangeal joint (MPJ) in horses. Their results showed that there is no apparent clinical benefit of repeated intra-articular administration of MSCs at a 1-month interval in horses with MPJ OA when set against the effect of a single injection [93].

### 6.2. Tendon and Ligament Injury

The injuries of the ligament–tendon apparatus are one of the most frequently observed injuries in orthopedics in dogs, cats, and horses. A large proportion of the cases concern animals taking an active part in sport [103,104,116]. Most often, in medical and veterinary practice, we deal with acute, traumatic rupture of a ligament or tendon, chronic abuse of tendon structures, or degenerative tendinopathy [117,118]. The degenerative causes of tendinopathies include senile changes, arthropathies of the immune background, and anatomical disorders [119]. Damage to a ligament or tendon, depending on its location and function, may be accompanied by severe pain and swelling in the surrounding tissues or a joint, as well as synovitis. Depending on the type of injury and the damaged tissue, biomechanical limitations in the movement of a given structure are also often observed. A prime example is damage to the cranial cruciate ligament (CrCL) in a dog, where during an attempt to take a step and load the pelvic limb, tibial craniocaudal subluxation and excessive medial rotation of the tibia occur. Due to the above-described situation, the animal experiences significant pain and discomfort, which is manifested by lameness of various degrees [104,120].

The low regenerative potential of the ligaments and tendons results in a prolonged and often limited healing process and, as a result, the animal does not regain its full fitness. The cascade of injury healing is associated with a low intensity of metabolic processes, which is largely influenced by a small number of cellular elements and scarce vascular supply [90,116,121,122]. Immediately after injury, in addition to the primary inflammatory response typical of healing, this process is characterized by progressive fibrosis of damaged tissues and is more often referred to as repair than regeneration. As a result, the formation of specific scar tissue is observed, which, compared to the primary tendon tissue, is characterized by a low organization of ECM, excessive tissue stiffness, and reduced flexibility. The formation of a scar significantly reduces the biomechanical functionality of a given structure and predisposes it to further injuries in the future [105].

The methods of conservative treatment of ligament and tendon injuries are very often pharmacologically supported through the use of systemic or local anti-inflammatory drugs and dietary supplements. For several years, these animals, at the time of deciding to start conservative treatment, are simultaneously subjected to physical therapy. Unfortunately, due to the severely limited regenerative abilities of the described tissues, the only effective type of treatment in most cases is surgical intervention [103,104,123,124,125].

The ideal therapy should therefore aim to regenerate the correct structure of the tendon or ligament. The idea of using tissue-engineered materials and MSCs was introduced as an alternative to the traditional approach, as it represents a potential tool for better tissue regeneration. Cell therapy with MSCs is aimed at restoring collagen fibers and restoring the normal activity of the tendons with a lower risk of recurrence [103,126].

In the beginning, the source of stem cells that were injected into the damaged tendon was bone marrow [127]. However, administering large volumes of bone marrow may worsen the situation due to the disruption of remaining intact tendon tissue [128]. Currently, in treating tendon disease in horses, there are two methods known using MSCs. In the first one, the isolated and multiplied bone marrow-derived MSCs are used, and in the second one, adipose-derived nucleated cell (ADNC) fractions or AD-MSCs after their multiplication are applied [129]. The impact of ADNC was examined in an experimental study with collagenase-induced tendinitis in horses. Ultrasonographic, gross, and histologic examination revealed an improvement in structural organization and a reduction of inflammation in the ADNC-treated tendons compared to the controls six weeks after ADNC injection [130]. The efficacy of BM-MSCs in healing tendon damages is correlated with the amount of injected cells. It was observed that a cell quantity of less than 1 × 10^6^ was insufficient for tendon healing [109]. Research with the use of 1 × 10^7^ BM-MSCs in healing a naturally occurring superficial digital flexor tendon damage (SDFT) showed the advantage of BM-MSCs therapy over the control group (saline) in the means of biomechanical (reduced stiffness), histological (lower scores), and compositional (lower GAG content) properties [131]. The distant clinical observations also seem promising. The racehorse group demonstrated a success rate of 90% following MSCs treatment of SDFT lesions. These horses successfully returned to their previous level of competition without re-injury for more than 2 years, while in the non-MSCs-treated control group, re-injury occurred in all horses after a median time of 7 months [109]. Another meaningful factor that has an impact on treatment efficacy is the implantation moment. It is suggested that the optimal time for MSCs implantation is 1–2 months after injury, after a suitable granulation bed has formed and before fibrosis starts to dominate [132]. The results of treating tendon injuries with MSCs are promising both in clinical and experimental studies. The improvement was observed in clinical, ultrasound, and post-mortem examination [130,133]. Romero et al. [105] experimentally evaluated the healing process of surgically damaged superficial digital flexor tendons in horses, injecting autologous bone marrow, adipose tissue derived mesenchymal stem cells, and platelet-rich plasma into the tendon. Their results showed that a clear beneficial effect was elicited by all treatments compared to the control group (lactated Ringer’s solution). Although differences among treatments were relatively small, BM-MSCs resulted in a better outcome than PRP and AD-MSCs [105]. Furthermore, the use of AD-MSCs and PRP use for the therapy of experimentally induced tendonitis prevented the tendon lesion progression. Ultrasound examination and histopathological evaluation showed much better collagen fibers organization as well as decreased cell inflammation at the site, induced with collagenase gel in the superficial digital flexor tendon in comparison to the control group (phosphate buffered saline —PBS) [108]. Beneficial treatment effects were demonstrated while injecting BM-MSCs into the tendon in naturally occurring tendinopathy in horses. During the research, the normalization of biomechanical, morphological, and structural parameters of the injured tendon was found after the BM-MSCs treatment in comparison to the control (saline) [131]. In horses, while treating suspensory ligament (SL) and SDFT, allogeneic peripheral blood mesenchymal stem cells were also used in allogeneic platelet-rich plasma. Two-year-long observations showed that injecting them into the tendon statistically lowered the re-injury rate 2 years after treatment significantly (*p* < 0.0001) in comparison to conventional therapies [103].

Retrospective studies in dogs proved that the stem cells preparations have positive effects in treating both partial cranial cruciate ligament tears [104] and supraspinatus tendinopathy [106,107]. The use of bone marrow aspirate concentrate (BMAC) or adipose-derived progenitor cells (ADPC) with PRP shows promise for the treatment of early partial CCL tears in dogs. The arthroscopic evaluation of partial cranial cruciate ligament tears conducted in 13 dogs 90 days after the injection of stem cells with PRP showed, in nine dogs, a fully intact CCL with marked neovascularization and a normal fiber pattern was found with all previous regions of disruption healed [104]. Similar promising results in a retrospective study were found using both BMAC with PRP [106], ADPC, and PRP in the treatment of supraspinatus tendinopathy [107].

In Table 2, selected animal studies with the results of tendinopathies and ligaments damage treatment in dogs and horses are presented.

## 7. Summary

Regenerative medicine and the use of stem cells are currently very popular in the world of science, which results from their possible application and the effectiveness of their activity in damaged organs and tissues. Researchers and clinicians from around the world are constantly working on the development of perfectly safe and effective protocols for the treatment of a given disease entity with the use of MSCs [1]. The effectiveness of the preparations used, the improvement of the patient’s clinical condition, as well as their long-term positive therapeutic effects have been proven many times in the studies. The noteworthy studies presented above in the field of diseases of the musculoskeletal system confirm that MSC therapy can be successfully used as an alternative to classic methods of treating orthopedic diseases [11,12]. The positive results of these studies encourage the further search for even better and more effective methods in regenerative medicine [1].

The discovery and development of methods of obtaining MSCs in the adult organism allowed for their clinical application both in humans and in animals [11,12,134]. One of the problematic topics in cell therapy is the optimal source of material isolation. The MSCs used in most studies were derived from bone marrow (BM-MSCs) or adipose tissue (AD-MSCs). The procedures for collecting the material and isolating individual cell lines from these tissues are now refined, but it is possible that there is still a better, yet undiscovered source and methods for isolating MSCs from tissues. The best example of that is the new approach relating to the possibility of obtaining cells with multipotent capacity from adipose tissue with a new medical device, that guarantee manipulation in a sterile device, without in vitro amplification and with a direct inoculation in the patient [135].

Unfortunately, so far, no standard protocols have been established regarding the effective dose of cells, the frequency of their application, or the method of administration in a specific disease entity. It is also necessary to pay attention to factors such as the donor’s age, sex, or past diseases. These factors can have a significant impact on the quality and quantity of isolated MSCs in material taken from a given source.

Another ethical aspect continues to be the use of autologous and allogeneic cells. It has been assumed that the use of preparations containing autologous MSCs is safer, but their acquisition and isolation is more complicated, longer in duration, and more expensive for the animal caregiver. An alternative is allogeneic cells derived from another individual of the same species. Unfortunately, there are still many unknowns regarding the use of these cells, such as their immunogenicity or the potential to trigger an immune response in the recipient’s organism.

Ideally, stem cells with total or pluripotent characteristics that can differentiate into any type of cell type in the body would be used. Unfortunately, their clinical use has been impossible so far. It is related to, inter alia, the fact that these cells are not fully controlled when administered to the patient, which can lead to their uncontrolled differentiation. This is confirmed by the results of studies carried out on animals, in which neoplastic changes were observed after the administration of ESCs [7]. In addition, their acquisition and use in the case of people raises ethical concerns [7,22]. Stem cells with multipotent properties when administered to a patient will not cause uncontrolled growth. This has been confirmed in both experimental and clinical studies [26,71,83]. Furthermore, their source is adult organisms, so their use does not raise ethical concerns. All this means that adult stem cells are often used in the regeneration and healing of body tissues, such as bone [2,136] or cartilage [12,13,134]. As mentioned above, their use in an auto-genital system requires their prior collection and isolation from the treated patient [11,20,134]. Therefore, it is necessary to perform an additional surgical procedure, which is associated with additional trauma for the patient, and carries an increased risk of intra- and perioperative infections.

Regenerative medicine is a dynamically developing field of human and veterinary medicine. It is an alternative option for the treatment methods used so far, not only for diseases within the musculoskeletal system but also for organ diseases or skin injuries. Despite numerous discoveries and positive therapeutic effects, further research is still needed, which will enable the standardization of protocols and the possibility of using MSCs in the therapy of a given disease entity. Researchers may soon be able to obtain satisfactory results, giving hope in the treatment of diseases that have so far been considered incurable. In recent years, MSC therapy is slowly becoming the gold standard in the treatment of musculoskeletal diseases. This is evidenced by numerous laboratory studies, as well as clinical patients who have been successfully treated with MSCs.

## Figures and Tables

**Table 1 biomolecules-11-01141-t001:** A collection of selected papers describing the use of MSCs in the treatment of OA in animals. Abbreviations: OA —osteoarthritis, AD-MSCs—adipose derived mesenchymal stem cells, PBS—phosphate buffered saline, PRGF—plasma rich in growth factors, PRP—platelet rich plasma, HA—hyaluronic acid, UCB-MSCs—umbilical cord blood mesenchymal stem cells, SVF—stromal vascular fraction, BM-MSCs—bone marrow derived mesenchymal stem cells, PB-MSCs—peripheral blood derived mesenchymal stem cells.

Disease	Cell Therapy and Type of Injection	Species	Number of Test Animals	Control Group	Number of Control Animals	Observation Period	Effect of Therapy
OA hip joint [40]	autologous AD-MSCs, 4.2–5 × 10^6^ cells, single intraarticular injection	dog	Group A (18 animals in total divided in groups)	yes, injection of placebo (PBS)	Group B	90 d	Positive therapeutic outcome of applied therapy.
OA elbow joint [95]	autologous AD-MSCs 3–5 × 10^6^ viable cells, single intraarticular injection	dog	14	lack	0	30, 60, 90, 180 d	Positive therapeutic outcome of applied therapy.
Degenerative fetlock joint disease [96]	allogeneic chondrogenic induced MSCs 2 × 10^6^, single intraarticular injection	horse	50	yes, injection of placebo	25	3, 6, 12, 18 w	Significant improvement in motor skills in the group of animals treated with MSCs.
OA hip joint [41]	AD-MSCs 30 × 10^6^ versus plasma rich in growth factors (PRGF), single intraarticular injection	dog	18	yes, injection of PRGF	17	1, 3, 6 m	Significant improvement in motor skills and the abolition of pain sensation in both groups.
OA elbow joint [13]	autologous AD-MSCs 3–5 × 10^6^ cells, single intraarticular injection with addition of PRP or HA as a scaffolds	dog	4	lack	0	1, 4 w	Positive therapeutic outcome of applied therapy.
OA metacarpophalangeal joint [93]	allogeneic UCB-MSCs, 10 × 10^6^ cells, single and repeated intraarticular injections	horse	14	MSC1 group received MSCs in M0 and placebo in M1 (D-PBS).	14	1, 2, 6 m	Positive therapeutic outcome in both test groups.
OA elbow joint [94]	allogeneic UCB-MSCs 7 × 10^6^ cells, single intraarticular injections	dog	30	saline (0,9% NaCl) placebo	25	1, 3, 6 m	Reduction in symptoms related to OA in the group of dogs that received a single injection of UCB-MSCs.
OA hip joint [97]	autologous stromal vascular fraction (SVF) 2–5 × 10^6^ and allogeneic AD-MSCs 2–8 × 10^5^, single acupuncture point injections	dog	9	lack	0	7, 15, 30 d	Positive therapeutic outcome of applied therapy.
OA lower hock joint [42]	autologous AD-MSCs, 5 × 10^6^ cells, single intraarticular injection	horse	10	Horses in group II received betamethasone intraarticularly, animals from group III had only limited movement during observation.	6	30, 60, 90, 180 d	Long-term benefits of MSCs therapy have been observed, as opposed to the short-term effects of steroids.
OA hip joint [43]	autologous AD-MSCs, 30 × 10^6^ cells, single intraarticular injection	dog	8	yes	5	30, 60, 90, 180 d	Increase in functionality and less discomfort associated with pain sensation caused by OA were found.
OA medial femoral condyle defect [98]	autologous BM-MSCs 20 × 10^6^ with addition of HA	horse	10	In the control group, only the intra-articular injection of HA was performed in the opposite stifle joint.	10	12 m	No significant clinical improvement was found after the completion of the follow-up.
OA stifle joint [99]	allogeneic UCB-MSCs, 1 × 10^6^ cells, single intraarticular injection	dog	4	Yes, saline injection of the same volume as in the group of animals tested.	4	3, 7, 14, 28, 35 d	Positive therapeutic outcome of applied therapy.
OA stifle joint [100]	allogeneic AD-MSCs 10 × 10^6^ with the addition of PBS, multiple intraarticular injections	dog	6 dogs in group III MSCs with PBS and 6 dogs in group IV MSCs and PRP	Yes, intra-articular injection of PBS in the control group.	6 dogs in group I PBS and 6 dogs in group II PRP	1, 2, 3 m	Beneficial effect of the medicinal PRP and MSCs preparations were found.
OA spinal region [101]	Allogeneic SVF-MSCs, 2 × 10^6^ cells per 1 kg of body mass, single intravenous injection	dog	10	Yes, blood samples were taken from the control animals to measure the level of VEGF.	10	2, 8, 24 w	Positive therapeutic outcome of applied therapy.
OA stifle, fetlock, pastern and coffin joints [102]	allogeneic PB-MSCs (chondrogenic induced and native) with PRP, single intraarticular injection	horse	165 (stifle n = 30, fetlock n = 58, pastern n = 34, coffin n = 43)	lack	0	6, 18 w	Positive therapeutic outcome of applied therapy.

**Table 2 biomolecules-11-01141-t002:** A collection of selected papers describing the use of MSCs in the treatment of injuries of the tendo-ligamentous apparatus in animals. Abbreviations: CrCl —cranial cruciate ligament, BM-MSCs—bone marrow derived mesenchymal stem cells, BMAC—bone marrow aspirate concentrate, AD-MSCs—adipose derived mesenchymal stem cells, ADPC—adipose-derived progenitor cells, PRP —platelet rich plasma, SL—suspensory ligament, SDFT—superficial digital flexor tendon, PBS —phosphate buffered saline.

Disease	Cell therapy and Type of Injection	Species	Number of Test Animals	Control Group	Number of Control Animals	Observation Period	Effect of Therapy
CrCL partial rupture [104]	autologous BMAC, ADPC with the addition of PRP, single intraarticular injection	dog	36 (19 dogs received BM-MSCs and 17 AD-MSCs)	lack	0	90 d	Promising clinical use of tissue-engineered products were observed.
Supraspinatus tendinopathy [106]	autologous BM-MSCs with PRP 1:1 ratio, ultrasound-guided single intratendionous injection	dog	41	lack	0	45, 90 d	Posttreatment ultrasound examination showed improvement in the structure of the tendon.
Supraspinatus tendinopathy [107]	autologous AD-MSCs 5 × 10^6^ cells/mL with PRP, ultrasound-guided single intratendinous injection	dog	55	lack	0	30, 60, 90 d	The applied MSCs therapy seems to be promising in comparison to conservative treatment.
SL and SDFT lesion [103]	allogeneic PB-MSCs 2–3 × 10^6^ cells with PRP, ultrasound-guided single intralesional injection	horse	104 (SL lesion in 68 individuals and SDFT in 36 individuals)	lack	0	6, 12 w, another survey 12 and 24 m after treatment	Positive therapeutic outcome of applied therapy.
SDFT lesion [108]	AD-MSCs 10 × 10^6^ cells with platelet concentrate, ultrasound-guided intralesional injection	horse	4	yes, PBS was used in the control group	4	every 2 weeks up to 16 w	Positive therapeutic outcome of applied therapy.
SDFT tendinopathy [31]	autologous BM-MSCs 10 × 10^6^ cells, ultrasound-guided intralesional injection	horse	141	lack	0	2 y follow up	The study showed that MSCs implantation is safe and appears to reduce the risk of recurrent tendon injury.
SDFT lesion [109]	autologous BM-MSCs (range 0.6–31.2 × 10^6^ cells), local or systemic injections	horse	11	yes	15	1, 3, 6 m	In 9 out of 11 horses, a significant improvement in movement and structure of the injured tendon (USG) was observed.
SDFT moderate to marked lesions [131]	autologous BM-MSCs 10 × 10^6^ cells, ultrasound-guided intra-tendinous injection	horse	6	yes, analogous injection of placebo (saline)	6	6 m	Cell therapy with autologous MSCs brought significant benefits over untreated tendon injuries in the control group.
Surgically induced SDFT lesion [105]	autologous BM-MSCs 20 × 10^6^ cells, AD-MSCs 20 × 10^6^ cells or PRP, single ultrasound-guided intralesional injection	horse	12 (24 tendons in the thoracic limbs were used for the study)	yes, 6 subjects received an injection directly into the tendon from the lactated Ringer	6 subjects received an injection directly into the tendon from the lactated Ringer	2, 6, 10, 20, 45 w	Favorable final effect was noticed in all groups of tested animals compared to the control group.

## Data Availability

The data presented in this study are available on request from the corresponding author.

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
