# Peer review of "The Role of Mesenchymal Stem Cells (MSCs) in Veterinary Medicine and Their Use in Musculoskeletal Disorders"

_biomolecules, 2021, doi:10.3390/biom11081141_

Round 1
Reviewer 1 Report
In this review the authors collect the studies on the use of stem cells for musculoskeletal disorder therapys. The topic is very interesting for esearchers in this field. The authors should give more prominence to this topic and summarize some paragraphs on mesenchymal cells. For example paragraphs 2, 3 and 4 could be summarized in a single paragraph. Perhaps the paragraphs concerning treatments for musculoskeletal disorders should be discussed more instead.
Author Response
Dear Reviewer,
we thank you for the sound and very helpful critique of the submitted article entitled: 'The Role of Mesenchymal Stem Cells (MSCs) in Veterinary Medicine and Their Use in Musculoskeletal Disorders". After reviewing the comments, we have improved the article and are sending it back to you for the re-evaluation. You will find it below

Reviewer 2 Report
The manuscript entitled "The Role of Mesenchymal Stem Cells (MSC) in Veterinary Medicine and their Use in Musculoskeletal Disorders" is focused on a topic of paramount interest in veterinary medicine and there is a strong linking with human field.
In general, the manuscript needs revisions with a particular regard to the structure of the review and to the topics discussed.
The review does not improve knowledge about MSCs use in veterinary medicine. The manuscript lacks of a whole and dedicated approach to the veterinary field.
In particular, it seems that there is not a clear scheme in the construction of the paper.
Title and abstract
The acronym MSC should be added with the “s”, as a plural voice.
Line 14 what does it mean “satisfactory effect”? The word effect is not clear.
Line 18 “organs”? The sentence is not clear
The abstract needs a revision regarding the aim of the paper and what the authors want to discuss.
Introduction
The chapter do not indicate the authors’ aim. It would be useful focusing the attention on veterinary medicine and the application of a regenerative approach in this specific field.
Line 28-29. Too general and not so accurate. It is necessary a clarification.
Division of MCSs
It would useful introducing the topic with any lines in which the authors describe how they want organize the chapter. How do they want to
Line 52: the “life of the body” are not the correct words.
From line 55 to line 83 bibliographic references are completely missing
Line 78-80: the sentence is not clear. It should be re-written
Line 86-89: too general. It seems that MSCs are able to heal any pathology or organ damage. It is necessary to better address the application field.
In general, from line 84 to line 95 the topic should be better organized and discussed.
Line 97: a mistake “in vintro”. “body”? It would be better to use the word tissue or organ.
Tissue Sources of MSCs
Line 127: it would be useful to add a reference at the end of the sentence.
Line 136-137: the sentence is not clear. What do the authors mean?
Line 147: It is important to underline that cells can be passed for a maximum of 4 times.
Autologous and Allogenic MSCs
Line190-191: “therapeutic therapy”?
Line 211: “cell lines”? Maybe, it would be better to talk about a batch not a cell line. In this specific case, if the authors want to describe the possibility to store cells in a biobanknig facility, they should consider the quality control, such as sterility, adventitious agents contamination, etc, that it is necessary to perform.
Immunomodulatory properties of MSCs
Line 116: “more and more” here and “more and more” at line 229
The chapter is very poor of information. It is necessary to improve and delve it. A little introduction of the concept that the authors want to consider. What is MHC class I, what is its specific function? Why it is important for the MSCs action?, etc…..
Application of stem cells in veterinary medicine
This chapter is just a list of the possible use of MSCs in veterinary medicine, but it is necessary to focus the attention on the topic of main interest.
Line 241: The punctuation should be revised, because the sentence is not clear.
Musculoskeletal System Disorders and Injuries
Line 260: what are the “adjacent tissues”?
Line 262-264 “injuries” and “injury” few words after. Traffic accident? Maybe this is can be true for a cat or dog, but it is more difficult for a horse.
If it was authors’ intention to use this chapter as an introduction, there should be some reference to the following chapters.
Degenerative Joint Disease
In this chapter, the authors should better explain table 1. In fact, it is very confused. It is necessary a legend of the acronyms used. The number of animals should be describe in a separate column as well the number of animals in control group. There are too many information in the column “effect of therapy”. It is necessary to summarize them.
Tendon and Ligament Injury
In this chapter there is a description of the main causes regarding tendon and ligament injury, but the part relating to the use of MScs for the treatment of this pathology is missing. There just few lines at the end of the chapter and a reference to table 2. The whole chapter should be revised.
Table 2 the same as table 1.
Summary
Line 338-347: lack of reference
The authors did not mentioned the new approach relating to the possibility of obtaining cells with multipotent capacity from adipose tissue with new medical device, that guarantee manipulation in a sterile device, without in vitro amplification and with a direct inoculation in the patient.
Author Response
Dear Reviewer,
we thank you for the sound and very helpful critique of the submitted article entitled: 'The Role of Mesenchymal Stem Cells (MSCs) in Veterinary Medicine and Their Use in Musculoskeletal Disorders". After reviewing the comments, we have improved the article and are sending it back to you for the re-evaluation. You will find it below.

Reviewer 3 Report
General comments
First of all, I would like to underline the fact that this paper is the result of a very large and tedious work of bibliography. The authors had to do a lot of synthesis work.
But in my opinion the major limit of this work is that by wanting to synthesize a lot of work on, what is more, on different species, many approximations, shortcomings, or missing information, appear making the scientific interest of the article debatable. In the same way I think that with so many bibliographical references it was difficult for the authors to analyze them all well (example of misinformation in the specific comments).
One of missing information is the current presence in veterinary medicine of cell therapy products that have received marketing authorization from European Medecines Agency (2 in horses and one in dogs).
Specific comment
Chapter 3. Tissue sources of MSCS
- Lines 141-144, authors write “Unfortunately, collecting the material is associated with a highly invasive procedure performed under general anesthesia and the risk of postoperative complications, such as infection or bleeding (in horses, even pneumothorax and hemothorax when collecting material from the sternum area) [32].”, but conclusions of the authors cited in the reference 32 are completely the opposite : “The most suitable sternal biopsy site was at the 4th or 5th sternebra. The surgical procedure was easy to perform and well tolerated by the horses, and adequate samples were obtained on the first attempt. The only complications were incisional edema in all horses and wound drainage in 1 horse. Conclusions—Sternal bone biopsy may be successfully performed in standing horses. Clinical Relevance—The sternum is an accessible site for cancellous bone biopsy specimens in standing horses.” To complete on this topic the authors should add a more recent publication on the bone marrow aspiration using a Jamshidi needle which is less invasive (Optimisation of bone marrow aspiration from the equine sternum for the safe recovery of mesenchymal stem cells. Kasashima et al. Equine Vet Journal 2013 43 (3) 288-294).
- Lines 166-174. In this part I think it could be interesting to add the reference “Culture and characterization of equine peripheral blood mesenchymal stroma cells”. Spass et al. The veterinary Journal (2013) 107-113.
- Line 189-191. “….using these cells in therapeutic therapy” what does it mean?
Chapter 4. Autologous and Allogenic MSCs
It is true that the tolerance of allogeneic cells is still being debated, however, as mentioned above, two allogeneic cell therapy products have a marketing authorization in horses and one in dogs, showing that tolerance is acceptable. To complete this statement, the authors could also quote the following article: “Bertoni et al. Intra-articular injection of 2 different dosages of autologous and allogeneic bone marrow-and umbilical cord-derived mesenchymal stem cells triggers a variable inflammatory response of the fetlock joint on 12 sound experimental horses. Stem Cell International (2019) 9431894.”
Chapter 6. Application of Stem cells in veterinary medicine
I find that in this part the authors synthesize to the extreme the use of stem cells in veterinary medicine, with little hierarchy of the most frequent indications. It would be necessary for example to specify the most frequent uses according to the species.
Chapter 7. Musculoskeletal System Disorders and Injuries
Same remark as above, the authors wanting to synthesize, certain general assertions cannot be applied to all species. For example lines 262-263 it is inadequate to consider that “very often injuries of the musculoskeletal system occur as a result of an injury or a history of a traffic accident”. It is probably true for dogs but not for horses.
Chapters 8 and 9
As mentioned before, there are approximations in the will to write generalities common to all species. Example lines 322-321: warming compresses are not recommended for conservative management of tendonitis in horses.
And, unfortunately I find these 2 chapters, it lacking an analysis from the authors. The tables only list the studies without much details or prioritization of the studies according to their power.
Conclusion
To conclude, I find this article lacking in precision. This is probably inherent to the author's desire to synthesize a lot of data from different species. The problem is that in the end it is difficult for the reader to get something very relevant for his veterinary practice.
Moreover, I find that this article is very close to an already published one (quoted by the authors), and that there is no real contribution to it (reference 29 “Stem cells in veterinary Medecine-current state and treatment options.Voga et al. Frontiers in Veterinary Science 2020”).
For all these reasons the article does not seem to me, as it stands, suitable for publication.
Author Response

(The authors gave the same response as above.)

Reviewer 4 Report
- Summary- This review paper seeks to describe and compare current approaches to mesenchymal stem cell-based therapies in veterinary medicine. Different sources and modes of introduction are discussed. In addition, the review summarizes recent research reports that have utilized stem cell therapy for a range of veterinary medical conditions.
- Broad comments Overall, this review may be of use for readers interested in a general appreciation of stem cell therapy in the veterinary context. Sometimes, it seems that the authors lose sight of the audience for this article (eg, the ethical issues that are sited several times in the article relate primarily to human stem cell therapies, not animal). With regard to the literature review, it seems a more focused review of studies with control groups would be more helpful. Without control groups, it seems that any purported stem-cell mediated improvement would be suspect. For such a shortened list, perhaps an expanded description of magnitude and statistical significance of effects would be more helpful than the rather thin description currently offered in this review.
- Specific comments
- Quite a few grammatical/stylistic issues throughout the article.
- Xenogeneic should be more clearly defined (line 198)
- Cost, as well as duration of processing steps should be addressed as a limitation of autologous approaches (line 206)
- Should address the potential problem of immunogenicity with xenogeneic transplants. (Line 231)
- I think that the section "Immunomodulatory Properties of MSCs" (lines 232-239) could easily be removed without detriment to the review. A new section that describes what stem cells are thought to do once introduced into their new location would be helpful.
- Should reference relevant studies found in the Table at the end of section 7 (line271)
- Need to address more than if using stem cells is "popular" (line 338)--rather, effectiveness and feasiblility is crucial. A more careful, critical analysis of these aspects would be useful.
- The discussion of bioderivatives and microvesicles seemed surprising at the very end of the article (line 394). This needs to be either moved earlier in the body of the article, or removed if not described more sufficiently.
Author Response

(The authors gave the same response as above.)

Round 2
Reviewer 2 Report
In general, the authors followed the suggestions and this revised version is clearer and the chapters have been improved in topic.
Introduction:
The authors have addressed in a more suitable way the chapter, focusing the aim of the paper on only two species: dog and horse.
Line 30-31 “In the most simple way” It is not grammatically correct
Line 51-54 The last sentence evidences some mistakes: i.e. “different forms”, try with another word.
Division of MCSs
Line 74-76 The sentence is not clear.
Line 92-95 There is a mistake in the construction of the sentence.
Autologous and allogenic MSCs
Line 241-244 This sentence lacks a sense of fulfilment in this position. Maybe it would be the conclusion of the chapter.
Immunomodulatory properties of MSCs
The authors have improved the information in the chapter as requested.
Line 289-290 “Nevertheless the risk of disease transmission from donor to recipient has to be kept in mind”, for this reason it is important to underline the requirement to schedule a specific and systematic plan of control of viral, bacterial, fungal potential contamination for each species.
Musculoskeletal System Disorders and Injuries and Examples of Stem Cell Application in Veterinary Medicine
The title is very long: is it possible to shorten it? Maybe it would be useful to focus the attention using less words.
Degenerative Joint Disease
The chapter has been improved and better addressed.
Table 1 was corrected for the acronyms and two columns were added to describe the number of animals involved in each study. Nonetheless, the column “effect of therapy” remains difficult to follow: in my opinion, there are too much information. It would be better to mention only the positive or negative effects during the treatment.
Tendon and Ligament Injury
The chapter has been revised and improved. The section added on line 433 to 487 requires a second overview for grammatical mistakes.
Table 2: the same as table 1.
Summary
The authors followed the suggestions and improved the information.
Author Response
Dear Reviewer,
we thank you for the sound and very helpful critique of the submitted article entitled: “The Role of Mesenchymal Stem Cells (MSCs) in Veterinary Medicine and Their Use in Musculoskeletal Disorders". After reviewing the comments, we have improved the article and are sending it back to you for the re-evaluation. You will find it below.
Review 2
In general, the authors followed the suggestions and this revised version is clearer and the chapters have been improved in topic.
Introduction:
The authors have addressed in a more suitable way the chapter, focusing the aim of the paper on only two species: dog and horse.
Line 30-31 “In the most simple way” It is not grammatically correct – has been corrected.
Line 51-54 The last sentence evidences some mistakes: i.e. “different forms”, try with another word. - has been corrected.
Division of MCSs
Line 74-76 The sentence is not clear. – has been corrected.
Line 92-95 There is a mistake in the construction of the sentence. – has been corrected.
Autologous and allogenic MSCs
Line 241-244 This sentence lacks a sense of fulfilment in this position. Maybe it would be the conclusion of the chapter. – has been corrected.
Immunomodulatory properties of MSCs
The authors have improved the information in the chapter as requested.
Line 289-290 “Nevertheless the risk of disease transmission from donor to recipient has to be kept in mind”, for this reason it is important to underline the requirement to schedule a specific and systematic plan of control of viral, bacterial, fungal potential contamination for each species. – this suggestion has been added
Musculoskeletal System Disorders and Injuries and Examples of Stem Cell Application in Veterinary Medicine
The title is very long: is it possible to shorten it? Maybe it would be useful to focus the attention using less words. - has been corrected.
Degenerative Joint Disease
The chapter has been improved and better addressed.
Table 1 was corrected for the acronyms and two columns were added to describe the number of animals involved in each study. Nonetheless, the column “effect of therapy” remains difficult to follow: in my opinion, there are too much information. It would be better to mention only the positive or negative effects during the treatment. - has been corrected.
Tendon and Ligament Injury
The chapter has been revised and improved. The section added on line 433 to 487 requires a second overview for grammatical mistakes. - has been corrected.
Table 2: the same as table 1. - has been corrected.
Summary
The authors followed the suggestions and improved the information.
